# Waste to Carbon: Densification of Torrefied Refuse-Derived Fuel

**Andrzej Białowiec** [1,*] , **Monika Micuda** [1] **and Jacek A. Koziel** [2]

[1] Faculty of Life Sciences and Technology, Wrocław University of Environmental and Life Sciences, C.K. Norwida 25, 50-375 Wrocław, Poland; micuda.monika@gmail.com

[2] Department of Agricultural and Biosystems Engineering, Iowa State University, Ames, IA 50011, USA; koziel@iastate.edu

\* Correspondence: andrzej.bialowiec@upwr.edu.pl; Tel.: +48-71-320-5973

**Abstract:** In this work, for the first time, the feasibility of obtaining carbonized refuse-derived fuel (CRDF) pelletization from municipal solid waste (MSW) was shown. Production of CRDF by torrefaction of MSW could be the future of recycling technology. The objective was to determine the applied pressure needed to produce CRDF pellets with compressive strength (CS) comparable to conventional biomass pellets. Also, the hypothesis that a binder (water glass (WG)) applied to CRDF as a coating can improve CS was tested. The pelletizing was based on the lab-scale production of CRDF pellets with pressure ranging from 8.5 MPa to 76.2 MPa. The resulting CS pellets increased from 0.06 MPa to 3.44 MPa with applied pelletizing pressure up to the threshold of 50.8 MPa, above which it did not significantly improve ($p < 0.05$). It was found that the addition of 10% WG to 50.8 MPa CRDF pellets or coating them with WG did not significantly improve the CS ($p < 0.05$). It was possible to produce durable pellets from CRDF. The CS was comparable to pine pellets. This research advances the concept of energy recovery from MSW, particularly by providing practical information on densification of CRDF originating from the torrefaction of the flammable fraction of MSW–refuse-derived fuel. Modification of CRDF through pelletization is proposed as preparation of lower volume fuel with projected lower costs of its storage and transportation and for a wider adoption of this technology.

**Keywords:** energy recovery; biochar pellets; torrefaction; circular economy; compressive strength; municipal waste; energy densification; waste to carbon; recycling; RDF

---

## 1. Introduction

Demographic growth and economic development cause considerable challenges for the contemporary global economy. Among them, one of the most important is the growing energy demand and the amount of municipal waste generated. According to the forecasts for 2040 by the International Energy Agency, the demand for electricity will increase by 30% [1]. Similarly, the waste management sector will continue to grow. According to research by the Organization for Economic Co-operation and Development (OECD), an increase in national income by 1% increases the amount of municipal waste by 0.69% [2]. Although the circular economy based on waste recycling and zero-waste trends are gaining support, the growing population and higher living standards result in more waste. The World Bank report states that by 2025, the volume of waste will increase by 2.2 billion tons a year worldwide [3].

A synergistic solution to waste recycling and an alternative source of fuel is the product of thermal treatment of municipal solid waste (MSW), which is carbonized refuse-derived fuel (CRDF) [4]. It is a type of biochar produced from the combustible fraction of municipal waste. It is a product

that can be a renewable low-emission fuel. Its fuel properties, such as the lower heating value (LHV) ranging from 19.6 MJ·kg$^{-1}$ to 25.3 MJ·kg$^{-1}$, compete with conventional energy generation solutions [4]. In addition, the higher heating value (HHV) of biochar, including CRDF, depending on the substrate used, can reach up to 35 MJ·kg$^{-1}$ [5]. This HHV is comparable with energy content in different types of coal, such as hard coal (HHV > 23.9 MJ·kg$^{-1}$), non-agglomerating highly volatile coals (17.4 < HHV < 23.9 MJ·kg$^{-1}$), or lignite (HHV < 17.4 MJ·kg$^{-1}$) [6]. Also, biochar is characterized by high energy density, hydrophobicity, improved abrasiveness and low ash content [7].

The transformation of MSW into CRDF allows for solving the problem of waste storage and disposal, and the CRDF produced can be a fully-fledged renewable fuel [8–10]. Initial economic evaluation of MSW torrefaction has been published by Stępień et al. [11] where some basic calculations of heat demand for the process were determined. Authors concluded that the heat demand for drying and torrefaction of MSW is ~ 1.27 GJ·Mg$^{-1}$. Assuming the heat utilization rate of 90%, the chemical energy introduced with fuel into a boiler is ~ 1.41 GJ·Mg$^{-1}$. Assuming the use of natural gas (~ $3/GJ; U.S. pricing), the total cost of drying and torrefaction is ~ $4.21·Mg$^{-1}$ of MSW. Obviously, that cost will differ for other markets due to fuel prices and fuel type. Additional operation costs were not included [11], but despite this, the comparison of MSW torrefaction costs with other MSW treatment methods shows that this is a competitive technique [12,13].

The production of CRDF can, therefore, be considered as a viable solution to the problem of management of emerging municipal waste. Thermal treatment reduces MSW volume and mass. For example, in Poland, organic waste accounts for over 80% of the total MSW [14]. Transforming it into CRDF via torrefaction or pyrolysis would allow for energy recovery and limiting the demand for disposal and storage. Thus, CRDF can be considered as a future-proof product. Besides the many advantages of CRDF, there are also challenges, e.g., effective bulk storage and transport. CRDF suffers from low bulk density and would therefore incur high transportation and storage costs. It has been reported that biomass densification can improve feedstock uniformity and enhance the handling and conveyance efficiencies [15,16].

The combination of pelletization with the thermal process, i.e., pyrolysis or torrefaction, improves the fuel properties of the product as well as the conditions of storage and transport. Solutions combining these processes into one technological line with continuous reactor operation are increasingly used [17]. The strength of the material is a parameter defining the limit value at which the body will be destroyed or irreversibly deformed. It depends on the type of material, shape, and the size of the sample, as well as the applied load and time [18]. Two basic static tests are used to determine the basic strength properties and deformation characteristics of the materials: compression and expansion. This makes it possible to determine the maximum compressive strength (CS) followed by the destruction of the material.

The CS of materials is determined by exerting axial thrusts on the analyzed samples using universal strength machines or hydraulic presses. The durability of pelletized biochar is very important due to its transport and storage. As mentioned earlier, the strength of the material, and thus the resistance to deformation, can be expressed using various parameters, e.g., abrasion resistance, brittleness, tensile strength or compressive strength.

The pelleting process increases the energy density, affects the unification of the material and gives it a regular shape. Also, pelleting affects the increase in grindability of the material [19]. This process is mainly used to improve fuel properties, as, in addition to increased energy density, humidity decreases, and a regular shape facilitates transport and subsequent burning in boilers [20]. Additives are often used during pelletizing for better compaction and binding of the material. However, binders play a major role in wood pellet characteristics. Additives improve pellet durability and physical quality, reduce the dust potential, improve pelleting efficiency and reduce energy costs [21]. The maximum content of 2% of additives is permitted in woody pellets [22]. No limitation exists for non-woody pellets [23], though it is a requirement to indicate the type and quantity used.

The most common additives are (1) water (used if the moisture content of the mixed material is too low) and (2) binders, which act as glue between the particles if the lignin content of the material is not enough to hold a pellet together. Lignin is a natural, optimal binder of biomass because it melts under the heat of the pellet mill [24]. However, if the lignin content of the biomass is low, it may be necessary to add other additives. One of the simplest binders is vegetable oil, but the most widely used substance overall is starch [25]. Obtained CRDF is similar to coal. Typical binders used for coal briquetting are starch, poly(vinyl acetate), molasses, sulfide liquors, carboxyl methylcellulose, tar, pitch, crude oil, clay, cement and sodium silicate [26].

One feasible additive is sodium silica, also known as water glass [27,28], which has been used for the preparation of briquettes from coal [26,29,30] with a ratio of up to 12%. Also, water glass has been proposed to be used as a coating film, making pellets waterproof [31]. Thus, analogous to coal, for CRDF pelletizing, the use of sodium silica as a binder and coating has been proposed in this research.

To date, there is no work on the structural modification of CRDF through pelletization as preparation for effective storage and transport due to the relatively low exploration of torrefaction of MSW. In this research, the authors propose to address the challenges above by the structural modification of CRDF through the densification of the material and the creation of so-called biochar pellets, which will be similar in mechanical properties to commercially available biomass pellets. To date, there are no published reports on pelletizing CRDF. Thus, the very practical questions to advance CRDF concept are: (1) the required pressure needed for pelletizing, (2) the determination of resulting CS, and (3) the need for pellet binders or coatings. Therefore, this research aimed to determine if:

(a)    the CS of CRDF pellets increases with the applied pressure during pelletization;
(b)    the CS of CRDF pellets increases with the addition of water glass as a binder and as a coating, and;
(c)    the CS of CRDF pellets is comparable to conventional biomass pellets.

## 2. Materials and Methods

### 2.1. CRDF Used in Experiments

The subject of the research was CRDF originating from the torrefaction of the flammable fraction of MSW—refuse-derived fuel. CRDF was produced in the torrefaction process at a temperature of 260 °C and a 50 min retention time in a batch reactor, according to the procedure used by Białowiec et al. [4]. The properties of the CRDF produced were tested in accordance with the international standard methods summarized in Table 1.

**Table 1.** Proximate and ultimate parameters of carbonized refuse-derived fuel (CRDF) used in this research, mean ± standard deviation, *n* = 3.

| Parameter | Standard Method | Unit | Mean ± SD |
|---|---|---|---|
| Moisture | [32] | % | 1.54 ± 0.36 |
| Loss on ignition | [33] | % *d.m.* | 79.9 ± 1.24 |
| C | [34] | % *d.m.* | 59.7 ± 1.63 |
| H | [34] | % *d.m.* | 6.07 ± 0.53 |
| N | [34] | % *d.m.* | 0.68 ± 0.02 |
| S | [35] | % *d.m.* | 0.17 ± 0.01 |
| O | [34] | % *d.m.* | 13.24 ± 0.92 |
| Cl | [35] | % *d.m.* | 0.80 ± 0.13 |
| Higher Heating Value | [36] | MJ·kg$^{-1}$ | 27.315 ± 1.183 |
| Lower Heating Value | [36] | MJ·kg$^{-1}$ | 25.953 ± 1.306 |
| Ash | [37] | % *d.m.* | 20.14 ± 1.24 |
| Bulk density | [38] | kg·m$^{-3}$ | 424.4 ± 150.2 |

Note: *d.m.* = dry matter.

### 2.2. Biomass Pellets (Reference) Used in the Experiment

For comparison of the CRDF-based results, biomass pellets (pine, lignocellulosic, and corn husks) were used as a reference. The pine pellets (from Kucharski Company, Wroclaw, Poland) were characterized by a 5.5% moisture content, 0.4% *d.m.* ash content, and a LHV of 17.64 MJ·kg$^{-1}$. The lignocellulosic pellets (Kaizen, Poland) were a 1:1 mixture of deciduous wood and softwood characterized by a 4.4% moisture content, 0.9% *d.m.* ash content, and a LHV of 17.90 MJ·kg$^{-1}$. The corn husk pellets (from Rabit Marcin Zajac, Poland) were characterized by a 4.8% moisture content, 1.2% *d.m.* ash content, and a LHV of 15.63 MJ·kg$^{-1}$. The diameter of the reference pellets ranged from 6.35 mm for pine to 8.89 mm for lignocellulosic, with height from 14.70 mm to 16.65 mm (Table 2). The cylindrical shape of the pellets from plant biomass was similar to the CRDF pellets produced.

**Table 2.** Dimensions of reference pellets from biomass, mean ± standard deviation, *n* = 5.

| Type of Biomass Pellet | Diameter, *d* | Height, *h* |
|---|---|---|
| | mm | mm |
| Pine pellet | 6.354 ± 0.054 | 16.654 ± 1.718 |
| Lignocellulosic pellet | 8.888 ± 0.081 | 16.206 ± 1.687 |
| Corn husk pellet | 8.410 ± 0.062 | 14.704 ± 1.410 |

### 2.3. Experimental Design

The CRDF densification and compressive strength was determined with the use of an Instron 5566 (Instron, Norwood, MA, USA) testing machine, and consisted of two phases:

1. Modification of the CRDF structure by applying controlled pressure (i.e., densification via pelletizing).
2. Stress tests of produced CRDF pellets and conventional biomass pellets (pine, lignocellulosic, corn husks which were used as a reference, Figure A1—Appendix A) to determine CS.

After the initial CRDF preparation, it was subjected to structure modification, as a completely randomized design experiment, through compaction (pelletization), resulting in pellets in three combinations (Table 3):

- combination I—CRDF using controlled pressure (from 8.5 MPa to 76.2 MPa with an interval of ~ 8.5 MPa)—checking the influence of just one factor—applied pressure—on CRDF pellet CS,
- combination II—CRDF with water glass as a binder (from 10% to 50%, with an interval of 10%, produced using the pelletization pressure of 50.8 MPa)—checking the influence of just one factor—applied dose of water glass to one chosen type of CRDF pellet—on CRDF pellet CS,
- combination III—CRDF pellets with water glass as a coating (produced using the pelletization pressure of 50.8 MPa)—checking the influence of just one factor—applied water glass as a coating agent of one chosen type of CRDF pellets—on CRDF pellet CS,
- reference biomass pellets.

  Pellets in all three experimental combinations and reference pellets were then subjected to CS tests.

**Table 3.** Experimental matrix for carbonized refuse-derived fuel (CRDF) densification, *n* = 5.

| Experimental Combination | Pelletizing Pressure Applied, MPa | The Ratio of Water Glass in Pellet, % |
|---|---|---|
| I | 76.2<br>67.6<br>59.2<br>50.8<br>42.3<br>33.9<br>25.4<br>17.0<br>8.5 | 0.0 |
| II | 50.8 | 10.0<br>20.0<br>30.0<br>40.0<br>50.0 |
| III | 50.8 | 8.1 |
| Pine pellets * | - | 0.0 |
| Lignocellulosic pellets * | - | 0.0 |
| Corn husk pellets * | - | 0.0 |

* used as a reference for comparisons.

### 2.4. Structural Modification of CRDF

After, the material was pre-treated and ground in the LMN-100 crushing mill (Testchem, Pszów, Poland) on a 1 mm diameter sieve, the structural modification of CRDF was carried out by compacting the material using the INSTRON 5566 (Instron, Norwood, MA, USA) testing machine at ambient temperature.

### 2.4.1. The Procedure of CRDF Structural Modification—Combination I

The CRDF pelletization took place in a metal sleeve (12 mm diameter × 48 mm length) (Figure A2—Appendix A), in which 2.0 g samples were compressed. The load and displacement were recorded with the BlueHill 2 software (Instron, Norwood, MA, USA). A detailed record of the compaction process in the form of compression pressure diagrams, with relaxation as a function of time, was obtained. The compression head moved at a rate of 0.3 mm·s$^{-1}$ with a subsequent material expansion (relaxation) time of 120 s. Five material samples were produced at each pressure tested. All prepared samples were weighed and measured with a caliper to determine the density of the CRDF pellets.

### 2.4.2. The Procedure of CRDF Structural Modification—Combination II

Combination II was based on the production of CRDF pellets with a water glass binder at 10%, 20%, 30%, 40%, and 50% (by weight) with a compressive pressure of 50.8 MPa. This pressure was determined to be practically sufficient (see Discussion). First, a mixture of 40 g of ground CRDF (grain size < 1 mm) was prepared with the percentage of water glass (Dragon Sod 145 type) with a density of 1.3 g·cm$^{-3}$ (Dragon Poland, Skawina, Poland). According to the specifications provided by the manufacturer, the water glass was 40% sodium silicate (CAS: 1344-09-8). The procedure for making pellets in this combination was identical to that described in Section 2.4.1.

### 2.4.3. The Procedure of CRDF Structural Modification—Combination III

Combination III was based on the coating of CRDF pellets in a solution of the same type of water glass mentioned in Section 2.4.2. CRDF pellet samples were produced with 50.8 MPa of pressure, as described in Section 2.4.1. Five samples were prepared that were measured and weighed to calculate the density. The sample was then immersed for 1 min in the water glass solution. The coated sample was allowed to air dry for 2 d. Dried samples were again measured and weighed.

### 2.5. The Compressive Strength Test of Pellets

The CS test was carried out with the use of the same testing machine and software as used for the production of the pellets. The CS tests were done according to standard procedure [39]. Due to differences in the physical dimensions of the tested pellets (Tables 2, 4 and 5), the applied force (N) to a cross-section of the pellet area (mm$^2$) was expressed in pressure units (MPa). The test consisted of compressing the CDRF pellets until destruction, recorded by the software as a crack. The crack point of the sample, and thus the maximum CS of the material, was marked in the form of a black triangle. In this test, the compression load was applied with a speed of 0.3 mm·s$^{-1}$. A detailed record of the pressure course in the form of compression pressure diagrams with relaxation as a function of time was obtained. The strength of 80 MPa was assumed as the safety limit. The CRDF pellets from combination I were limited to those produced using 17.0 MPa to 76.2 MPa because pellets produced with 8.5 MPa disintegrated, i.e., no durable pellet was obtained. The tests were carried out in five replicates for each type of modified CRDF, and for the biomass pellets. CS was estimated as the ratio of breaking load to the cross-section area of each pellet and expressed in MPa units.

### 2.6. Statistical Analyses

The obtained results were subjected to statistical verification. The descriptive statistics like mean values and standard deviation were determined. Differences between means were tested by performing an analysis of variance with the Tukey post-hoc test at a significance level of $p < 0.05$. Statistical analyzes were performed using the Statistica 12 software (StatSoft, Inc., TIBCO Software Inc. Palo Alto, CA, USA).

## 3. Results

### 3.1. Structural Modification of CRDF

#### 3.1.1. CRDF Pellets—Combination I

The compressive load over time for each sample type is illustrated in Figure A3 (Appendix A). Stress relaxation tests allowed us to determine the response of the material, i.e., a reduction of pressure over time as shown in Figure A3 (Appendix A). The presented changes in applied pressure as a function of time show a gradual increase in load, reaching the maximum value of a set point used to pelletize the CRDF. Then, an apparent decrease in the load values was measured during the stress relaxation phase (up to 120 s). After relaxation, the remaining pressures were reduced to 69.9 MPa (90.5%), 60.8 MPa (89.9%), 51.8 MPa (87.5%), 43.0 MPa (85.0%), 34.2 MPa (80.8%), 26.1 MPa (77.0%), 18.3 MPa (72.0%), 10.1 MPa (59.4%), and 5.1 MPa (60.0%), compared with the initial applied pressures ranging from 76.2 MPa to 8.5 MPa, respectively. The apparent trends in loads are the same for all samples; no significant deviations between successive repeats were observed. This observation is important for scale-up considerations and mass production of CRDF pellets.

It should be noted that at 8.5 MPa, the resulting sample was too loose, and the material could not be compacted to a pellet. The diameter of the obtained pellets was between 12.26 mm to 12.27 mm with no apparent correlation to the applied pressure. Changes were observed in the height of the pellets. The greater the pressure, the lower the height, ranging from 14.27 mm (76.2 MPa) to 16.95 mm (17.0 MPa). Thus, pellets produced with greater load resulted in a higher material

density of 1185.5 kg·m$^{-3}$ (76.2 MPa) to 999.9 kg·m$^{-3}$ (17.0 MPa) (Table 4). The visual assessment of the CRDF pellets revealed that the samples below 42.3 MPa were brittle, disintegrated and did not take on a perfectly cylindrical shape as in the case of pellets produced with greater pressure (Figure A1—Appendix A). Pellet integrity and shape is important for scale up to production and adoption of this technology.

**Table 4.** Dimensions, weight, and density of carbonized refuse-derived fuel (CRDF) pellets produced at pressures from 8.5 MPa to 76.2 MPa, and CRDF pellets produced with a pressure of 50.8 MPa with the addition of water glass content ranging from 10% to 50% as a binder, mean ± standard deviation, *n* = 5.

| Pellets Produced with Applied Pelletizing Pressure, MPa | The Ratio of Water Glass in Pellet | Diameter, *d* | Height, *h* | Weight, *m* | Volume, *V* | Density, $\rho$ |
|---|---|---|---|---|---|---|
| MPa | % | mm | mm | g | mm$^{-3}$ | kg·m$^{-3}$ |
| 76.2 | | 12,266 ± 0.022 | 14.266 ± 0.080 | 1.998 ± 0.004 | 1685.8 ± 15.2 | 1185.5 ± 11.6 |
| 67.6 | | 12.276 ± 0.030 | 14.354 ± 0.090 | 2.000 ± 0.002 | 1698.9 ± 9.4 | 1177.0 ± 6.8 |
| 59.2 | | 12.274 ± 0.017 | 14.332 ± 0.140 | 1.999 ± 0.001 | 1695.8 ± 17.6 | 1178.7 ± 13.0 |
| 50.8 | | 12.264 ± 0.009 | 14.513 ± 0.052 | 2.001 ± 0.002 | 1714.4 ± 4.5 | 1167.1 ± 2.7 |
| 42.3 | 0.0 | 12.264 ± 0.017 | 14.717 ± 0.040 | 1.997 ± 0.003 | 1738.5 ± 7.7 | 1149.0 ± 6.7 |
| 33.9 | | 12.266 ± 0.015 | 15.174 ± 0.089 | 1.999 ± 0.003 | 1793.1 ± 8.2 | 1114.8 ± 5.3 |
| 25.4 | | 12.272 ± 0.018 | 15.970 ± 0.110 | 1.999 ± 0.003 | 1889.0 ± 12.2 | 1058.1 ± 7.6 |
| 17.0 | | 12.240 ± 0.021 | 16.953 ± 0.130 | 1.991 ± 0.003 | 1994.8 ± 15.7 | 997.9 ± 8.6 |
| 8.5 | | - | - | - | - | - |
| 50.8 | 10 | 12.346 ± 0.021 | 14.474 ± 0.187 | 1.960 ± 0.015 | 1732.7 ± 19,2 | 1131.1 ± 19.6 |
| | 20 | 12.436 ± 0.035 | 14.288 ± 0.171 | 1.909 ± 0.012 | 1735.4 ± 16,9 | 1100.1 ± 92.0 |
| | 30 | 12.399 ± 0.025 | 13.931 ± 0.111 | 1.900 ± 0.012 | 1682.1 ± 16,8 | 1129.4 ± 12.7 |
| | 40 | 12.305 ± 0.058 | 13.465 ± 0.225 | 1.881 ± 0.040 | 1601.2 ± 34,1 | 1175.1 ± 22.9 |
| | 50 | 12.180 ± 0.118 | 12.905 ± 0.386 | 1.820 ± 0.090 | 1503.1 ± 26.3 | 1210.4 ± 51.9 |

### 3.1.2. CRDF Pellets with Glass Water as a Binder—Combination II

Combination II was based on the production of CRDF pellets with the addition of water glass as a binder with a ratio of 10% to 50%. Pellets produced at 50.8 MPa were used for experiments with water glass. The 50.8 MPa pellets were chosen as being produced with sufficient strength (which did not significantly improve at higher pressure, as discussed in detail in Section 3.2.1). In the case of CRDF pelletization with 40% and 50% binder content, resulting variability was much greater, and the process itself was more difficult to implement because water glass caused the CRDF to float in a metal sleeve. This variability can be observed in the load graphs of pellet formation (Figure A4—Appendix A). In the case of 50% water glass content, samples were non-homogeneous and very wet, which caused apparent spikes in the recorded load. This observation is important for scale-up considerations and mass production of CRDF pellets.

The course of the tests carried out until the end of the stress relaxation process (up to 120 s) indicates a decrease in the load values. After relaxation, the real pressures consisted of 90.1%, 93.1%, 91.3%, 85.6%, 82.7% of the initial value for CRDF (50.5 MPa) with a binder content of 10%, 20%, 30%, 40%, and 50%, respectively.

The diameter of the obtained pellets differed depending on the added water glass content and ranged from 12.18 mm for 50% glass to 12.44 mm for 20% (Table 4). In the case of pellet height, a decrease was observed with the addition of glass from 14.47 mm (10%) to 12.91 mm (50%). The differences are also measured in the mass of the sample, which is related to the water glass content. The density of CRDF pellets produced with the addition of water glass in the range of 10–30% was lower than in the case of those obtained without a binder. The increase of glass water content above 30% caused an increase of density to values exceeding the density of (50.8 MPa) CRDF pellets produced without a binder. Differences in the structure were observed using a visual assessment of the densified

material with water glass (Figure A1—Appendix A). Pellets with a lower water glass ratio were more compact and homogeneous compared with those with a 50% content.

### 3.1.3. CRDF Pellets Coated with Glass Water—Combination III

The last combination of the CRDF structure modification was the impregnation of the CRDF pellets by water glass. The obtained pellets increased their dimensions from 12.13 mm to 12.36 mm with a diameter of ~ 0.5 mm at height, and there was an increase in weight. The density of the samples was also higher (Table 5). During the coating process, the hydrophobic character of the CRDF could be seen, as the water glass did not soak into the pellet sample, and after the sample dried, a characteristic coat formed (Figure A1—Appendix A).

**Table 5.** Dimensions, weight, and density of carbonized refuse-derived fuel (CRDF) pellets (produced with 50.8 MPa pressure) coated with water glass, mean $\pm$ standard deviation, $n$ = 5.

| Type of Pellets | Diameter, $d$ | Height, $h$ | Weight, $m$ | Volume, $V$ | Density, $\rho$ |
|---|---|---|---|---|---|
| | mm | mm | g | mm$^3$ | kg·m$^{-3}$ |
| Before coating | 12.262 $\pm$ 0.008 | 14.525 $\pm$ 0.020 | 2.000 $\pm$ 0.000 | 1715.294 $\pm$ 2.993 | 1160.9 $\pm$ 1.9 |
| After coating | 12.356 $\pm$ 0.058 | 15.080 $\pm$ 0.142 | 2.176 $\pm$ 0.043 | 1808.122 $\pm$ 8.303 | 1203.7 $\pm$ 22.5 |
| Increase, % | 0.77 | 3.61 | 8.80 | 5.41 | 3.69 |

### 3.2. The Compressive Strength Tests

This research aimed to produce CRDF pellets, whose properties will be comparable to pellets from various biomass available on the market. Therefore, compression testing of both the CRDF pellets and pellets made of the lignocellulosic (reference) material was carried out.

### 3.2.1. The Compressive Strength of CRDF Pellets—Combination I

The results of the CS tests carried out for CRDF pellets produced at a variable pressure from 17.0 MPa to 76.2 MPa are presented in the form of response to compression loads (Figure 1), and comparative graph with analysis of variance (Figure 2).

High homogeneity of the material in the case of pellets compacted at pressures of 33.9 MPa to 76.2 MPa allowed us to achieve reproducible pellet properties (Figure 1) and no significant deviations between repetitions. Irregular test runs and larger deviations in the case of CRDF pellets obtained in the 17.0 MPa to 25.4 MPa range were observed. This is likely related to the greater heterogeneity of the material. The pellets produced at 17.0 MPa were loose and disintegrated when handled manually, hence the irregular and inconsistent pattern of sample cracks (marked with black triangles in Figure 1).

Due to the differences in dimensions and properties between individual samples, the most precise parameter illustrating the strength of the compressive material is the maximum CS expressed in MPa. An evident, significant ($p < 0.05$) decrease in strength with the compressive pressure of 3.94 MPa for the sample produced at 76.2 MPa to just 0.06 MPa for the 17.0 MPa sample is visible (Figure 2). There were small differences in the CS of only ~ 0.5 MPa for pellets produced with 50.8 MPa to 76.2 MPa, and they were not statistically significant ($p < 0.05$). The greatest drop in strength was observed for pellets produced within the 33.9 MPa to 50.8 MPa range. As mentioned earlier, the samples produced at 17.0 MPa were falling apart when handled manually, which is also visible in the graph (Figure 2), because the strength of this combination was only 0.06 MPa.

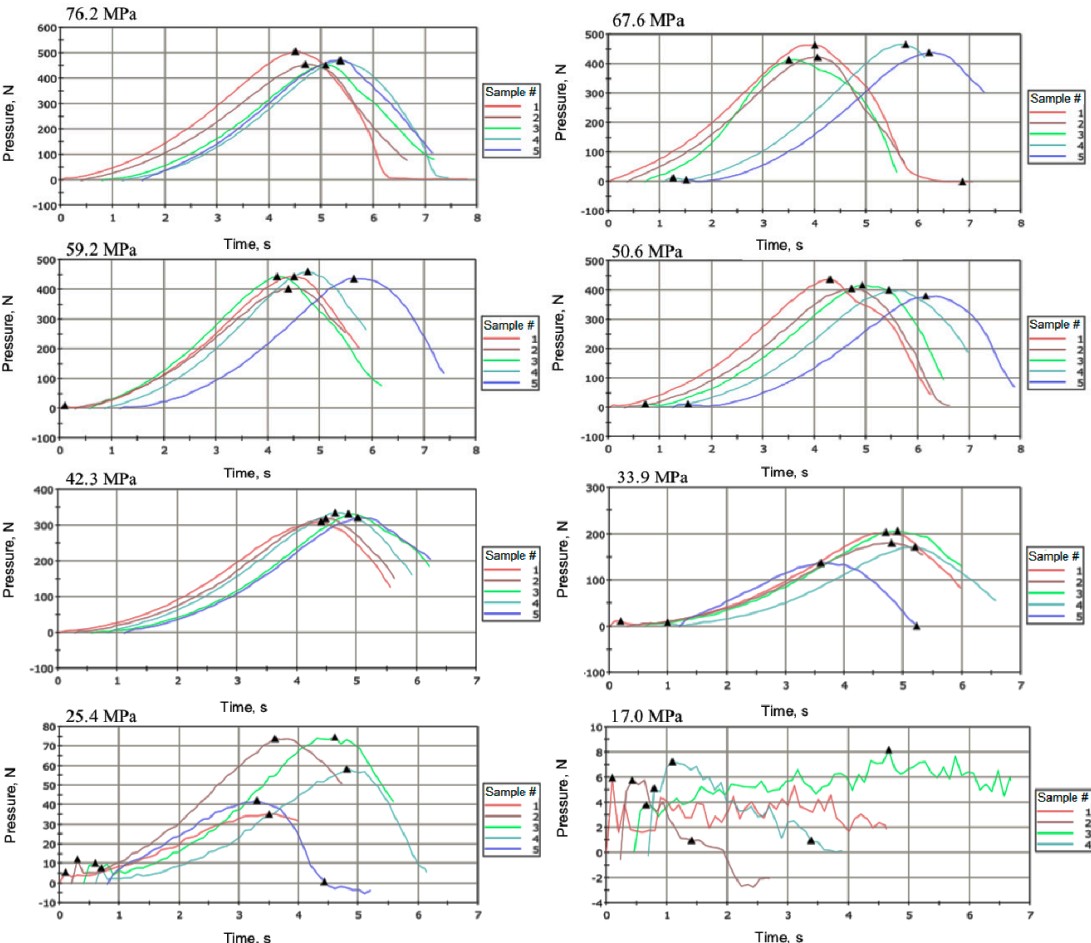

**Figure 1.** The response to the compressive load of carbonized refuse-derived fuel (CRDF) pellets produced with different pressures ranging from 17.0 MPa to 72.6 MPa. Black triangles signify the crack point of the pellet sample, and thus the maximum compressive strength of the densified material.

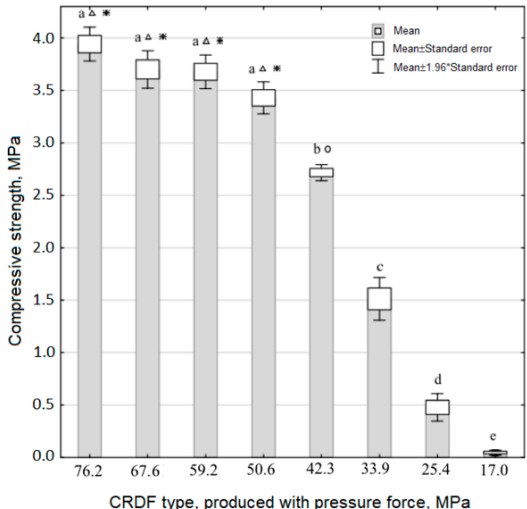

**Figure 2.** Compression strength of carbonized refuse-derived fuel (CRDF) pellets as a function of applied pressure used for pellet production—letter designations 'a–e' denote statistically significant ($p < 0.05$) differences between samples inside combination I, geometric markings indicate statistically significant ($p < 0.05$) differences between combinations I, II and III.

### 3.2.2. The Compressive Strength of CRDF Pellets with the Addition of Water Glass as a Binder—Combination II

The CS of CRDF pellets with the addition of water glass varied depending on the amount of binder added. A greater binder amount resulted in a greater heterogeneity of the material, and thus the response to load was associated with larger standard deviations between the individual samples (Figure 3). The lowest ($p < 0.05$) values are obtained with a 50% addition of water glass, thus the strength for this combination was generally lower. Only a 10% addition of water glass can provide similar ($p < 0.05$) strength to CRDF pellets obtained at 50.8 MPa.

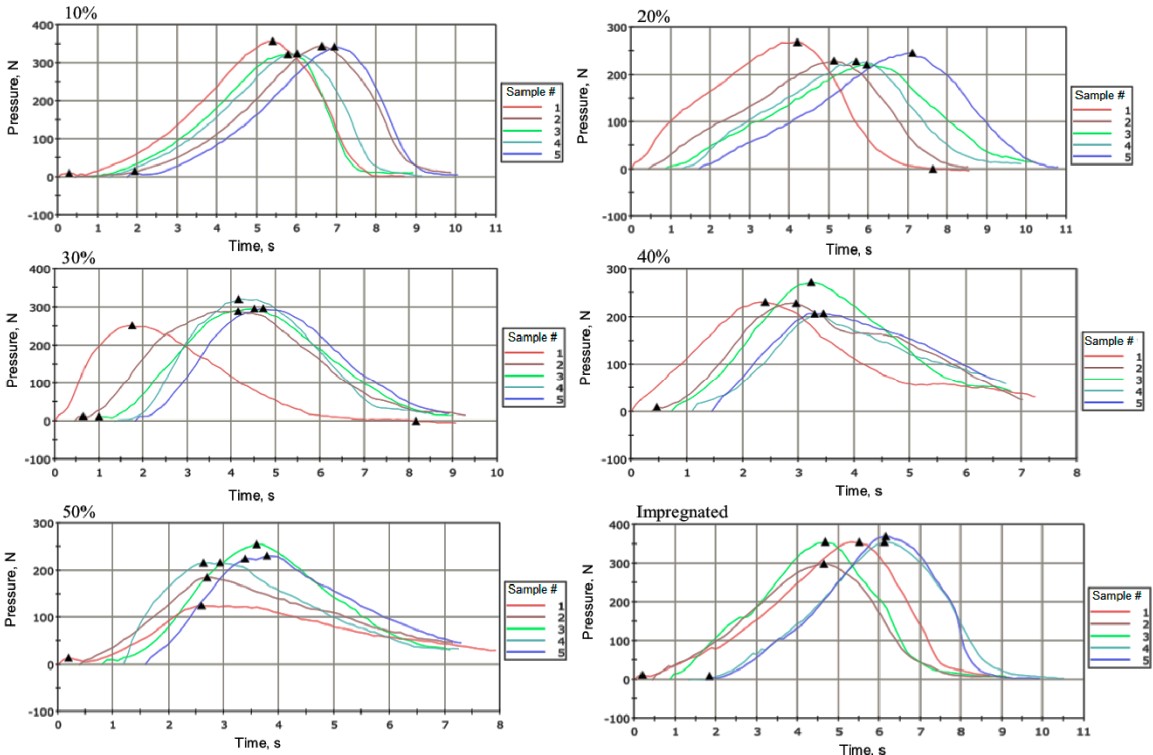

**Figure 3.** The response to the compressive load of carbonized refuse-derived fuel (CRDF) pellets (all produced at 50.8 MPa), with a 10% to 50% water glass content as a binder addition or with water glass as a coating. Black triangles signify the crack point of the pellet sample, and thus the maximum compressive strength of the densified material.

The maximum CS for CRDF pellets with the addition of water glass ranged from 2.84 MPa for a sample with 10% added binder to 0.7 MPa at 50% (Figure 4). The trend of significant ($p < 0.05$) decreasing strength in response to the added binder was observed. The only exception is the 30% addition of water glass, at which the sample reached 2.41 MPa CS, which was greater than pellets with 20% water glass addition (1.97 MPa). The share of water glass significantly ($p < 0.05$) reduced the CS of CRDF pellets.

### 3.2.3. The Compressive Strength of CRDF Pellets Coated with Water Glass—Combination III

In the case of pellets coated with water glass (Figure 3), there was a high reproducibility of the test. It should also be noted that the water glass in this combination formed a characteristic casing, which was not destroyed during the compression test. The resulting properties of the coated combination were similar ($p < 0.05$) to those with 10% glass. In the case of coated pellets, the CS was 2.89 MPa. This value is comparable to that achieved with pellets with 10% water glass addition (2.84 MPa—in this combination there were larger deviations) (Figure 4).

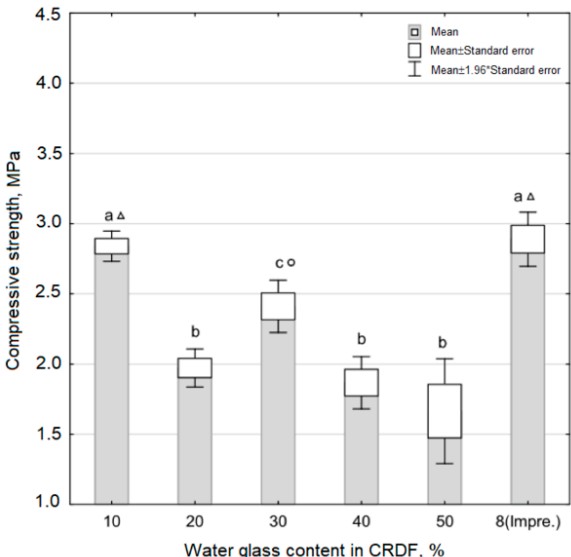

**Figure 4.** The compressive strength of carbonized refuse-derived fuel (CRDF) pellets produced at a pressure of 50.8 MPa depending on the water glass content in CRDF and coating of CRDF with water glass—letter designations 'a–e' denote statistically significant ($p < 0.05$) differences between samples inside combination I, geometric markings statistically significant ($p < 0.05$) differences between combinations I, II and III.

### 3.2.4. The Compressive Strength of (Reference) Plant Biomass Pellets

A heterogeneity of biological material was apparent. This was because individual pellets from the same type of plant biomass were of different heights (standard deviations in the range of 1.41–1.17 mm), which made it difficult to carry out strength tests. Therefore, the obtained characteristics (Figure 5) were associated with greater inherent variability. Large differences between individual pellets from plant biomass were also found in the CS test. Average CS ranged from 8.85 MPa for pellets from corn husk to 4.29 MPa for pine pellets (Figure 6). The average strength for biomass is ~ 6 MPa higher than in the case of CRDF pellets.

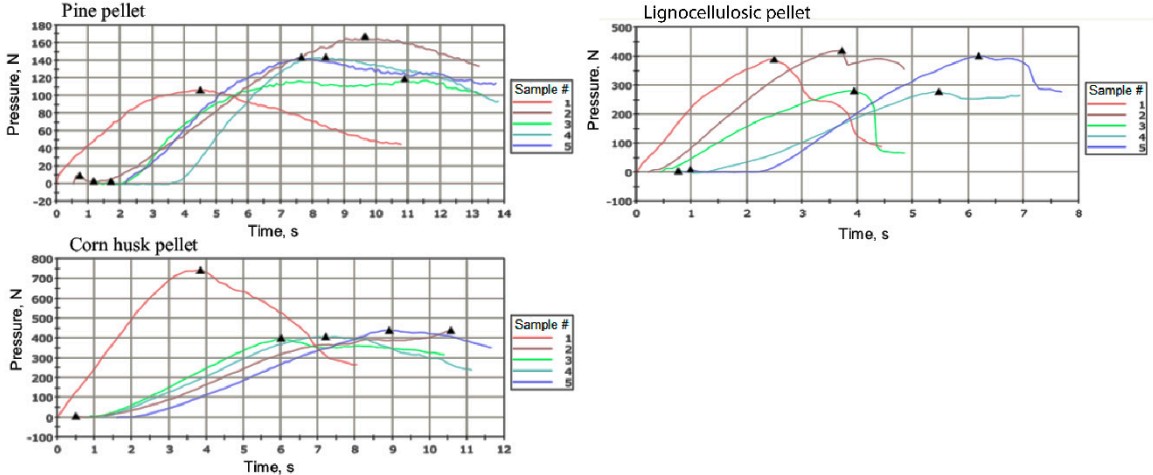

**Figure 5.** The response to the compressive load of pellets made of pine wood, lignocellulosic biomass, and corn husks. Black triangles signify the crack point of the pellet sample, and thus the maximum compressive strength of the densified material.

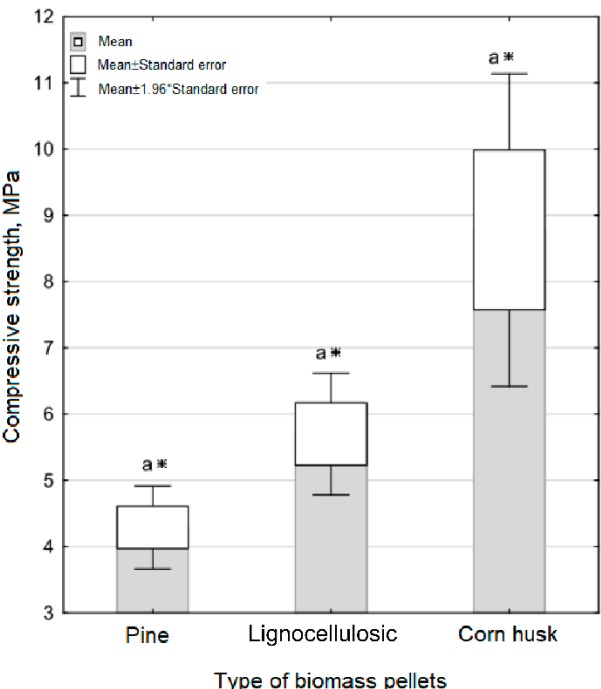

**Figure 6.** The compressive strength of pellets made of pine, lignocellulosic biomass, and corn husk—letter markings 'a–e' signify statistically significant ($p < 0.05$) differences between samples of pellets from plant biomass, geometrical markings indicate statistically significant ($p < 0.05$) differences between combinations I, II and III.

## 4. Discussion

In this work, for the first time, the feasibility of CRDF pelletization obtained from MSW was carried out. Previous work on biochar pelletization concerned products made of lignocellulosic biomass [15], and torrefied lignocellulosic biomass (woody tropical trees [16]) which in many cases did not have the status of waste. The analyzed CRDF from the torrefaction of municipal waste at 260 °C and 50 min of retention time was characterized by physicochemical properties similar to those described in the literature. CRDF of a LHV of 25.95 MJ·kg$^{-1}$ (Table 1) was similar to CRDF obtained in earlier studies [4] and to biochar from grass produced in a similar temperature range (250 °C to 350 °C) by [5], which had a calorific value of 25 MJ·kg$^{-1}$ to 30 MJ·kg$^{-1}$. The HHV of CRDF used in this experiment (27.315 MJ·kg$^{-1}$) could define it as hard coal (HHV > 23.9 MJ·kg$^{-1}$), according to the classification given by EUROSTAT [6]. The moisture content of the analyzed material (1.54%) was in the 1% to 6% range presented by Jakubiak and Kordylewski [19]. Small differences in properties could result from both different CRDF production parameters (temperatures, the residence time in the reactor), as well as from the high heterogeneity of municipal waste used for the production of CRDF.

Previously published research has shown that torrefaction/pyrolysis influences the mechanical strength of the biochar. Emmerich and Luengot [40] have shown that carbonized material can achieve similar or even better strength parameters than other materials, but it should be produced from a very durable material, e.g., the Brazilian native palm tree. Additionally, the pyrolysis temperature influence on the strength of the material has been reported [41]. It has been observed that up to a certain temperature (600 °C), the strength decreases due to the decrease in density caused by the ingress of volatiles and moisture, but after evaporation, there is an increase. Noumi et al. [42] conducted experiments on the strength of biochar from eucalyptus trees formed under different temperature conditions (350 °C and 600 °C), rates of temperature increase (1 °C and 5 °C min$^{-1}$) and pressure (2 bar and 6 bar) and concluded that the best strength parameters were associated with biochar produced at a higher temperature, shorter retention time and lower pressure. Additionally, the correlation of mechanical stability with the structure of biochar (porosity) and density was demonstrated.

In this research, it has been proven that densification via pelletization increases the mechanical strength of the biochar. Most previous work on solid fuel densification was done on different types of biomass. Biomass pellets are usually produced at pressures between 1.5 MPa and 300 MPa [43], and generally, higher pressures give more durable pellets [44]. Two studies using very low pressures (1.5 MPa) produced poorer pellets compared to standard pellets [45,46]. Higher pressures increase durability in cereal residues [47] and reduce pellet relaxation after formation [48]. A study on olive pruning residues found no difference in durability in pellets produced between 70 MPa and 175 MPa, although interactions between pressure and other factors suggested 170–180 MPa was optimal [49]. Another study suggested that only marginal improvements in durability could be achieved in beech and Scots pine above 250 MPa [50].

In this study, CRDF pellets were produced with pressure ranging from 8.5 MPa to 72.6 MPa, which generally falls in the range of pressures used for pelletizing biomass and confirms the durability increase with the increase of used pressure for pelletization, which is consistent with the results of densification of biochar from woody residues [44].

The pressure applied during pelletization at an industrial scale can be affected by a number of factors including the motor power, the rolling speed, the density of the feed, and the dimensions and material of the pellet channel [51]. Two studies suggest an interaction between pressure, temperature and moisture content [44,50], i.e., heat and moisture can ease the flow of material through the die, and therefore would need to be optimized for CRDF on a technical scale to ensure that the desired durability is achieved. Another factor may be the type of feedstock and pre-treatment procedure (e.g., application of torrefaction or pyrolysis).

The increase in material density is directly proportional to its strength [5]. It may be concluded that higher applied pressure result in higher sample density, which is related to the reduction of space in the biochar/biomass structure (lower porosity of the material). This relationship was described by Weber and Quicker [5]. However, the densification degree, or CS improvement degree, have limitations—threshold values above which the increase of applied pressure during pelletization does not increase the CS of pellets. In this research, this concept has been confirmed. The first (to date) CS analyses of CRDF pellets showed that the CS of compressed material produced with pressures over 50.8 MPa does not result in a significant improvement of CS: i.e., from 3.43 MPa for 50.8 MPa to 3.94 MPa for 76.2 MPa. Taking into account technological and economic considerations, it was decided that the pressure of 50.8 MPa was sufficient to obtain robust CRDF pellets. The strength of CRDF pellets obtained at 50.8 MPa statistically ($p < 0.05$) did not differ from those produced at higher pressure, and the compaction process itself was easier and therefore less expensive to implement. The same rationale was used to test the effects of water glass addition.

In the present study, it has been shown that the addition of water glass reduces the mechanical stability of the material. Only a 10% addition of the binder caused a reduction in strength by 0.59 MPa (from 3.43 MPa to 2.84 MPa). In the experiments carried out by Chinmayananda et al. [27], a slight improvement in CS was found with the use of water glass (ratio 30–35%) as a binding agent. Nevertheless, Chinmayananda et al. [27] did not specify the origin and properties of the biochar used in research. Thus, further investigation on the influence of the type of feedstock and process temperature on the durability of pellets with water glass should be carried out. In contrast, the strength of the investigated combination coated with water glass was similar to pellets with a 10% addition of water glass. However, authors do not recommend the use of water glass as a binder or as a coating material for CRDF pellets. Additionally, it was found that the water glass coating was troublesome due to the hydrophobic CRDF structure.

The obtained results of CS of CRDF pellets have been compared to biochar pellets obtained by other authors (Table 6). The CS is reported in units of pressure and force to enable comparisons between different studies. The necessary pressure-to-force conversion involved using the cross-sectional area of a pellet. The CS of most durable CRDF pellets without water glass, expressed in kN, were in the range of 0.405 kN to 0.465 kN, but the addition of or coating with water glass reduced the CS to 0.340–0.346 kN.

In comparison, pellets obtained from biochar produced from five different tropical trees had CS values in the wide range of 0.165 kN to 1.469 kN (Table 6) [16], which shows that the CS of CRDF pellets is located in the lower half of the values of biochar pellets from tree biomass. This indicates the strong influence of the type of feedstock on the final durability of the carbonized product.

**Table 6.** Comparison of compressive strength of carbonized refuse-derived fuel (CRDF) pellets with other biomass biochar pellets, and biomass pelletization.

| Feedstock for Biochar Production | Pyrolysis Conditions, °C; min | Additives for Pelletization | Compressive Strength, MPa (kN) | Source |
|---|---|---|---|---|
| **Biochar Pellets** | | | | |
| RDF | 260; 50 | - | 3.43 (0.405)–3.94 (0.465) | Present Study |
| | | Water glass (10%) | 2.84 (0.340) | |
| | | Water glass coating (8.09) | 2.89 (0.346) | |
| White cedar | | - | (0.323–0.929) | |
| Almendro | | - | 0.342–0.841) | Gaitán-Alvarez et al. [16] |
| Beechwood | 200–250; 8–12 | - | (0.165–0.583) | |
| Teak | | - | (0.251–0.620) | |
| Yemeri wood | | - | (0.624–1.469) | |
| **Biomass Pellets** | | | | |
| Pine sawdust | - | - | 4.29 (0.136) | Present Study |
| Lignocellulosic | - | - | 4.84 (0.300) | |
| Corn husk | - | - | 8.85 (0.491) | |
| Eucalyptus pellets | - | - | 5 | |
| Mixed wood pellets | - | - | 7 | |
| Miscanthus pellets | - | - | 7 | Williams et al. [52] |
| Sunflower pellets | - | - | 8 | |
| Steam exploded pellets | - | - | 15 | |
| Microwave pellets | - | - | 3 | |
| Hay | - | - | 3.29 | |
| Straw | - | - | 4.21 | Lisowski et al. [53] |
| Hay:Straw (1:1) | - | - | 3.74 | |
| Hay:Straw (1:1) | - | CaCO$_3$ (10%) | 4.76 | |

As a part of '*Waste to Carbon*' technology development, pellets with comparable strength properties to conventional biomass pellets should be achieved. This is needed to implement easily manageable, durable, highly calorific fuel to the market. The potential end-users have access to mature technology for pellet manufacturing, storage, handling, and utilization technology. On the other hand, carbonized (not densified) CRDF material has properties resembling powdery dust, raising concerns about safety (e.g., self-ignition), storage, transportation, handling, and utilization. Therefore, the CS of CRDF pellets was compared with the CS of biomass pellets available on the market. The tested lignocellulosic pellets and corn husk pellets showed better strength properties than CRDF pellets. Similar conclusions about higher CS associated with fibrous biomass have been reached in [5], i.e., that biochar produced from biomass has generally poorer strength compared with unprocessed biomass and coal. The CS of lignocellulosic and corn husk pellets was in a comparable (i.e., the same order of magnitude) range of values with the CS of CRDF pellets.

The tested CS of pine pellets was similar ($p < 0.05$) to the CS of CRDF pellets produced with the pressure of 50.8 MPa. Obtained CS values for CRDF pellets are comparable to this research and other studies on CS of biomass pellets (Table 6), which indicates that they could be competitive to biomass pellets on the market.

Based on Chinmayananda et al. [27], in experiments where pellet binders were used in the present study, we decided to verify the usefulness of water glass addition for the increase of CRDF pellet durability. It has been shown that as a result of compressing CRDF samples, the air was displaced from the inter-granular space, the particles were closer together and the grains connected together. Therefore, CRDF compressed at 76.2 MPa resulted in a higher sample density than the ones produced

at lower pressures. There are also visible differences in the volume of obtained pellets, which is also related to the porosity of the material. The higher compressive pressure caused a reduction of volume in the obtained sample. In the case of pellets with an added binder, the density of the material increased with the water glass addition. A similar observation was made in the case of structural modification involving the water glass coating. However, a bulk density analysis of (50.8 MPa) CRDF with and without the addition of water glass showed that water glass filling the spaces between the grains reduced the density of the material.

Finally, further investigation on smaller doses of water glass, and with other binding agents, could be pursued. It is recommended to conduct experiments on the influence of the addition of binding agents to CRDF on firing process conditions, properties of ash, and emitted gasses. The aforementioned test should be completed on both lab and technical scales for determination of binding mechanisms and interaction between molecules in the hydrophobic material, as well as for further evaluation of the technological, and economic feasibility of CRDF pelletization. Pelletization of the CRDF in this research was carried out on a small, lab scale testing machine. It is necessary to introduce a structure modification solution using production-scale pelletizing equipment to optimize implementation, verify performance, and to obtain data for economic analyses. This research has proven that CRDF pelletization and the production of durable CRDF is possible on a lab scale. The technology of pelletization is well known and may be easily used for CRDF pelletization. As the next step of RDF torrefaction technology development, a full-scale test is warranted to assess the energy demand for initial grinding and pelletization.

## 5. Conclusions

This research advances the concept of energy recovery from municipal solid waste (MSW), particularly by providing practical information on densification of carbonized refuse-derived fuel (CRDF) originating from the torrefaction of the flammable fraction of MSW. Modification of CRDF through densification (via pelletization) is proposed as preparation of lower volume fuel with projected lower costs of storage, transportation and utilization for a wider adoption of *'Waste to Carbon'* concept by a mature, pellet-based technology.

The following conclusions were made:

- It is possible to produce durable pellets from CRDF. This, in turn, improves the feasibility of adopting this technology for municipal waste recycling and energy recovery due to lower volume and projected lower costs of storage, transportation, and utilization.
- The 50.8 MPa pressure was the practical threshold value for CRDF densification. Further increases in pressure during pelletizing did not significantly increase the compressive strength (CS) of CRDF pellets.
- The CS of CRDF pellets is comparable to pine pellets.
- The addition of water glass binder reduces the CS of pellets. The water glass content should not exceed 10%.
- Coating CRDF pellets with water glass did not improve the CS.
- The use of other binders and coatings in general remains an open issue and research is warranted if practical durability issues arise with site-, waste-, and process-specific scale-up of the densification of CRDF.

It has been proven that CRDF pelletization and production of durable CRDF is possible on a lab scale. The technology of pelletization is well known and may be easily used for CRDF pelletization. As the next step of RDF torrefaction technology development, a full-scale test is warranted to assess the energy demand for initial grinding and pelletization.

**Author Contributions:** Conceptualization, A.B.; methodology, A.B., M.M.; formal analysis, A.B., M.M.; validation, A.B., M.M. and J.K.; investigation, M.M.; resources, A.B.; data curation, M.M., A.B.; writing—original draft preparation, A.B., M.M.; writing—review and editing, A.B., J.K.; visualization, A.B., M.M. and J.K.; supervision, A.B., J.K.

**Funding:** The authors would like to thank the Fulbright Foundation for funding the project titled "Research on pollutants emission from Carbonized Refuse Derived Fuel into the environment," completed at Iowa State University. In addition, this project was partially supported by the Iowa Agriculture and Home Economics Experiment Station, Ames, Iowa. Project no. IOW05400 (Animal Production Systems: Synthesis of Methods to Determine Triple Bottom Line Sustainability from Findings of Reductionist Research) sponsored by Hatch Act and State of Iowa funds.

**Acknowledgments:** The authors thank Krzysztof Lech for help with measurements on the Instron machine.

**Conflicts of Interest:** The authors declare no conflict of interest. The funders had no role in the design of the study; in the collection, analyses, or interpretation of data; in the writing of the manuscript, or in the decision to publish the results.

## Appendix A

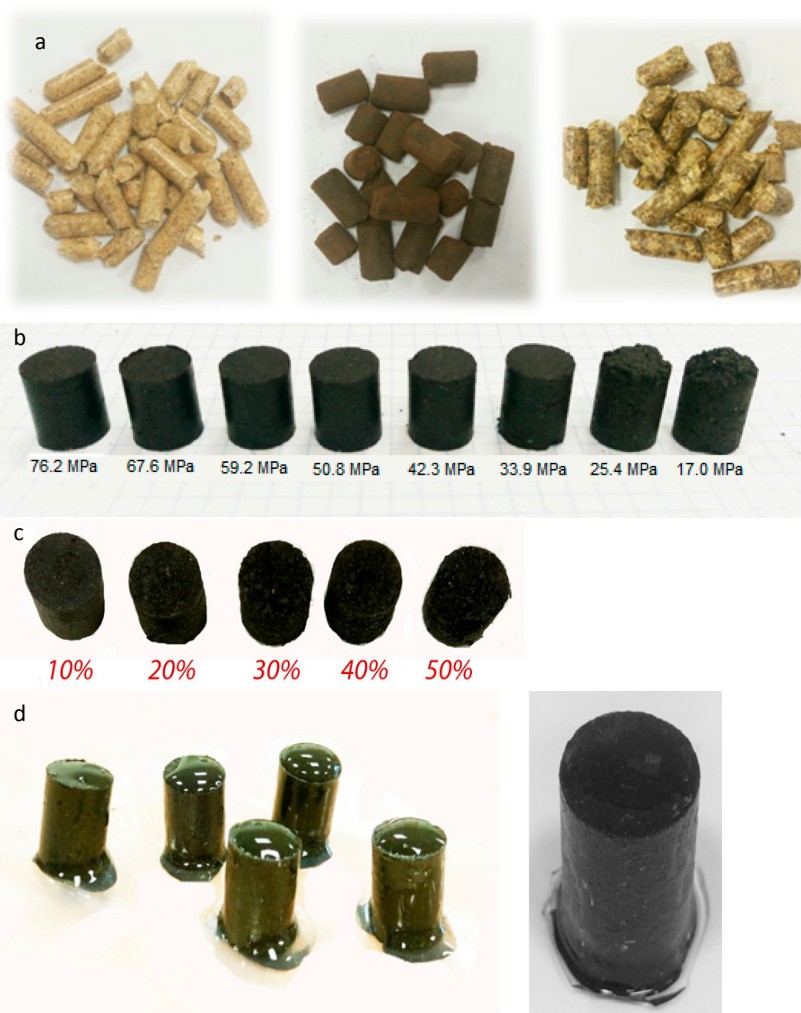

**Figure A1.** Pellets tested in the experiment: (**a**) reference pellets from biomass: pine (left), lignocellulosic (middle), from corn husks (right); (**b**) carbonized refuse-derived fuel (CRDF) pellets produced by applying pressure from 17.0 MPa to 76.2 MPa; (**c**) CRDF pellets produced with a pressure of 50.8 MPa and a variable proportion of water glass from 10% to 50%; (**d**) CRDF pellets produced with a pressure of 50.8 MPa impregnated with water glass: left: samples immediately after impregnation, right: sample after drying.

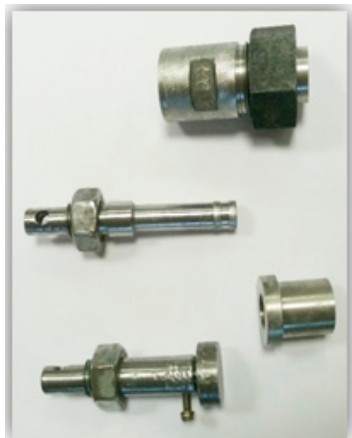

**Figure A2.** Metal fixtures and sleeves (right) used in the structural modification (pelletization) of carbonized refuse-derived fuel (CRDF).

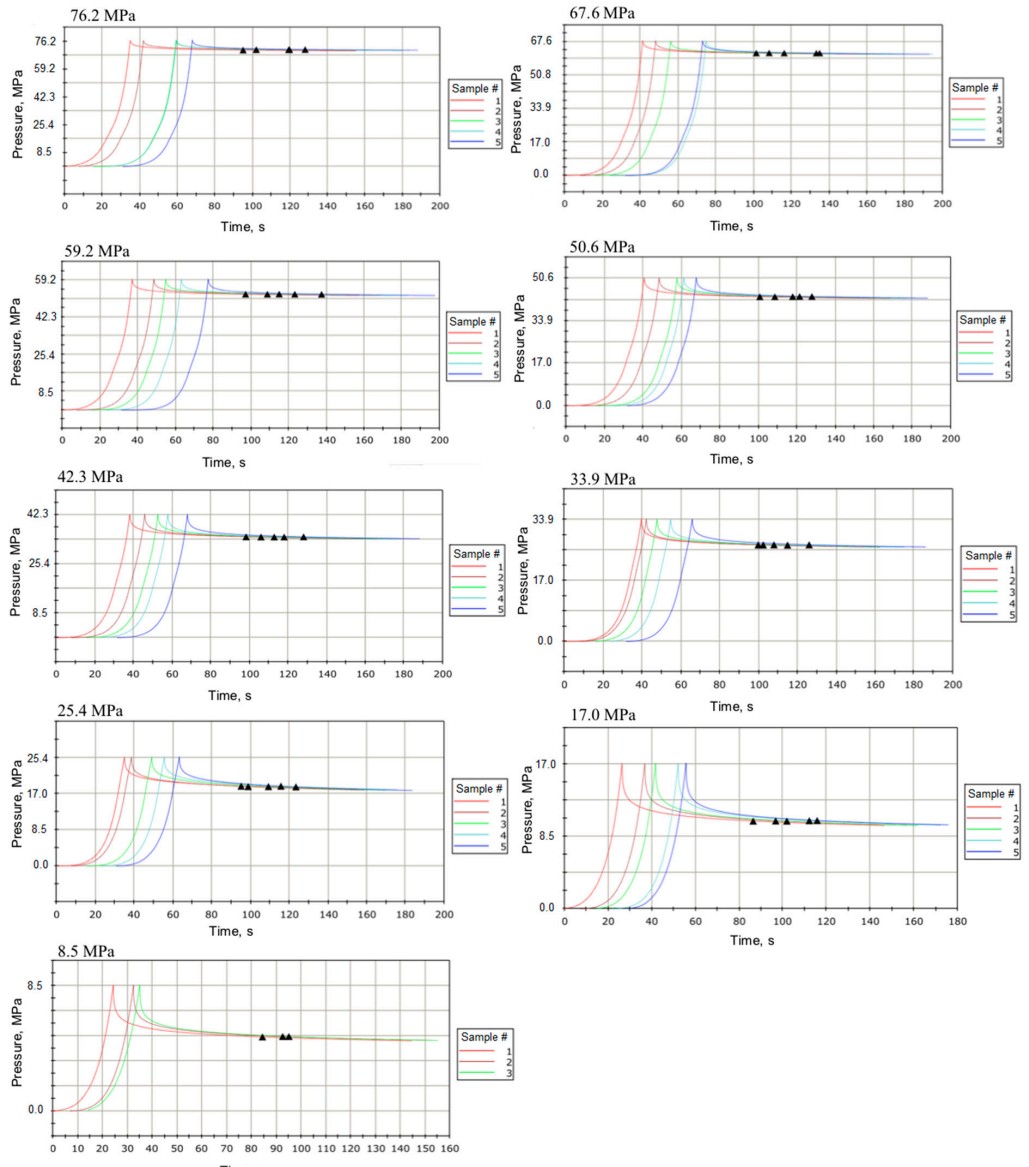

**Figure A3.** The densification (pelletization) of carbonized refuse-derived fuel (CRDF) with applied pressure varying from 8.5 MPa to 76.2 MPa.

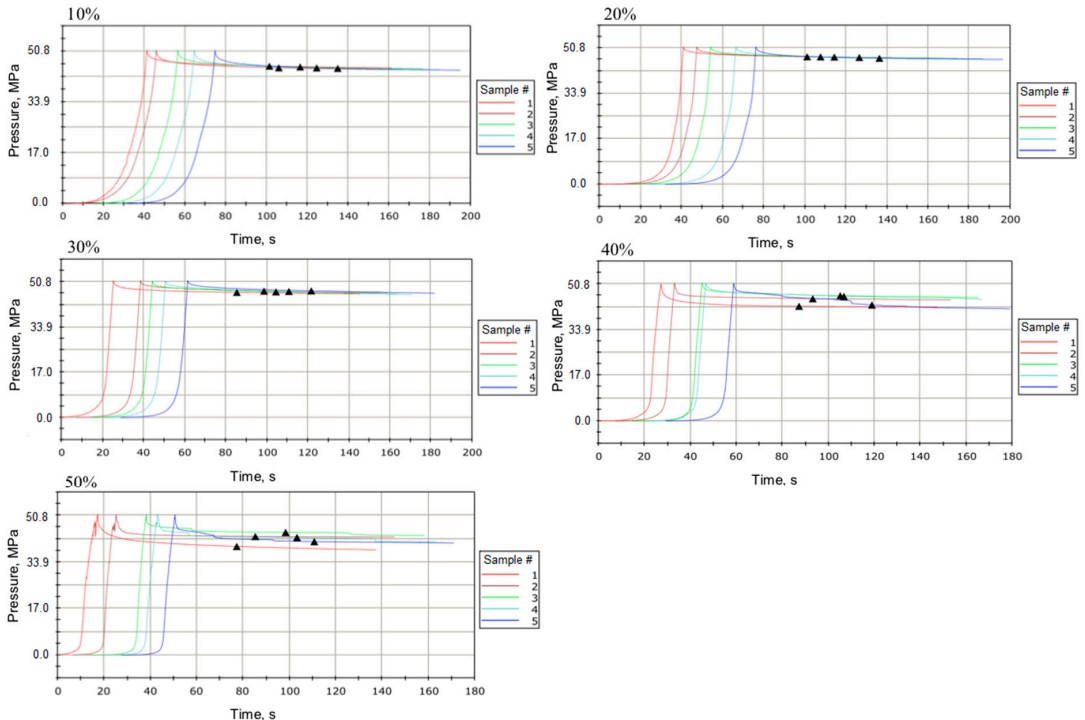

**Figure A4.** The densification (pelletization) of carbonized refuse-derived fuel (CRDF) with a pressure of 50.8 MPa, and with a water glass content ranging from 10% to 50%.

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
