# Peer review of "Waste to Carbon: Densification of Torrefied Refuse-Derived Fuel"

_energies, doi:10.3390/en11113233_

Round 1
Reviewer 1 Report
In general, this is a well-written manuscript. It includes several groups of experiments on the CRDF pelletization processes. The experimental processes are reliable, and the test results are interesting. However, there are some issues should be resolved before its publication:
(1) Some of the writing is overly complex and embellished, and will be difficult for non-native speakers. The author’s should consider re-wording some of these passages including: Line 11, Line 25 to Line 26, Line 34 to Line 35, and Line 37 to Line 38.
(2) Typos on Line 234. The “1.1.1” should be removed.
(3) This article uses biomass pellets as references. However, the dimensions of these biomass pellets are different from the CRDF pellets produced in this study. In Table 2, the diameters of these biomass pellets are from 6mm to 8 mm, and some of the heights exceed 18mm. But in Table 4, the CRDF pellets have diameters around 12mm, with heights approaching 17mm. This means that the CDRF pellets are up to 4 times the volume. Do the dimensional differences effect the CS test results? And what is the pelletizing pressure for the biomass pellets?
(4) The Discussion section is too long, and is only partly a discussion in the classical sense. A lot of the material is more a discussion of results from other papers rather than a focused assessment of test result presented in the manuscript. Part of this should move to the introduction at the start of the paper and the parts that really compare something should have the values put into one or more tables so that the values can be clearly compared and then the text can propose reasons for the measured differences. Also, putting some of the results on graphs along with results from prior work would make it much easier to follow arguments about trends in particular parameters.
(5) Along the same lines, parts of the discussion are repetitive. The authors should focus more on discussing the data presented in the results section and how various trends should, or should not, be interpreted.
Author Response
We would like to thank the reviewer for very helpful comments and suggestions. We responded to all comments in the attached file. In most cases, we followed them. We resubmit the corrected manuscript with track changes mode on. We submit the clear version of the revised manuscript with highlighted changes as reviewer 3 asked.

Reviewer 2 Report
The manuscript is written on densification of torrefied refuse derived fuel. It is written well with a reasonable transition of the content. The results are supported clearly by the data, and the graphs are clear and easy to understand. However, there are some missing information and technical errors that need to be addressed as follows:
1)What equipment and what models used to determine/measure properties of CRDF in Table 1? Should be included in the text under ‘Materials and Methods’.
2)In Table 3, variant should be replaced by ‘variable’ or ‘combination’.
3)Please replace ‘variant’ in the text by ‘variable’ or ‘combination’.
4)Line 120: Please put ‘The compressive strength was determined with use of Instron 5566’ instead of ‘The research was completed with the use of INSTRON 5566’
5)Line 120: How compressive strength of the pellets was measured? Which method was used?
6)Why water glass was used as the binder in such a high percentage (10%), while it is rich in ash and would reduce the energy content of the product?
7)What are the properties of water glass used in this study in terms of ash and particle size?
8)Line 150: If this measured by caliper then it would be unit density. Please revise this in the text and under results and Tables wherever applicable.
9)Line 156: is the value 1.3 unit density or bulk density of the water glass?
10)Line 157: Please revise g/cm to g/cm3.
11)Line 179: What was the experimental design? Was it a completely randomized design (CRD) or another type? Please explain and include in the text.
12)Table 4: Please replace bulk density with unit density.
Author Response

(The authors gave the same response as above.)

Reviewer 3 Report
Highlight changes in yellow in a next revision, please. No track changes.
Please add brief contextualization, see:
https://www.mdpi.com/journal/energies/instructions
Revise italics for “p”
End with practical implications (also in Conclusions, in order to enhance the study)
Keywords, I would better contextualize general terms
I would suggest removing personal expressions as “We”, etc
This should be the last paragraph relating what was addressed in the study, because then you “go back” to Introduction:
“In this research, we”
I would revise less used abbreviations: “a.k.a.”
Table 1
Revise references after “;”: “; [25, 26],”
Check all typos: “fuel, RDF).”
To references in tables, I would always add authors: direct references
Italics to variables: “Mean ± SD”; “d.m.”; see all parameter cases, as you do in Table 2, for example.
Always present units inside “()” for clarification
There is no interest in repeating the same information in a single column… Table 3, there are other ways…
Section 2.4: read…
Revise “First, the”
Statistical language needs further clarification: “2.6. Statistical analyses”
Terms like descriptive statistics and inferential statistics could help, etc…
Revise italics again: “p”
To me, it makes no sense to refer in details to supplementary information:
“Figure A1 (Appendix) illustrates”
Revise italics… “n=5.”
Revise typos “1.1.1. 3.1.3. CRDF pellets coated”
I feel there is a significant number of headings in the text and that difficult the perception of the work and results obtained
Figure 1: tiny content, not enough quality…
There must be no reference to other figures (and… that will come much later…) in captions…
“between variants I, II and III (Figure 4, 6)”
In the Discussion section, which is extensive the focus should be on the results of this work, although some comparison, reference to other works, may be made. I believe it is a mix of…
There are parts of the discussion that should be at the end of the conclusions (why must present quantitative data and should end with practical implications of such a novel (enhance originality/novelty then… ) study…:
“Finally, further investigation on smaller”
So, the two last sections could be revised in order to improve the relevance of the text
The conclusions section could start with a brief contextualization enhancing the study…
Overall comments: the text is comprehensive and enlightening.
Some changes in the number of headings (more cohesive text) and structure might contribute to improve it.
Author Response

(The authors gave the same response as above.)

Round 2
Reviewer 1 Report
manuscript is acceptable
Author Response
We would like to thank Reviewer #1 for significant comments. English language and style have been revised.
Reviewer 2 Report
The manuscript now is in good shape and is suggested for publication.
Author Response
We would like to thank Reviewer #2 for significant comments. English language and style have been revised.
Reviewer 3 Report
Highlight changes in yellow in a next revision, please. No track changes.
Consider comments in the entire text.
Where are they then?
“All changes are highlighted yellow in the revised manuscript. ”
“1)Please add brief contextualization, see:
(…)
Author’s response:
It has been revised.”
I was asking this in abstract too:
“3)End with practical implications (also in Conclusions, in order to enhance the study)”
Not enough, in my view and please no dot use abbreviations alone…
“4)Keywords, I would better contextualize general terms
Author’s response:
We revised the list of keywords to contextualize this research more effectively. We have replaced the keyword ‘water glass’ with ‘waste management’ as more related to practice”
I believe captions must be always self-explanatory. Authors could better contextualize “Properties” in “Table 1. Properties of CRDF”values
[Or “Table 3. Experimental matrix, n=5.”]
See all
Table 3, considerer using always the same number of digits because of “8.09”, consistence…
It was not, see non-italicized term in “2.6. Statistical analyses”:
“17)Revise italics again: “p”
Author’s response:
It has been revised.”
Yes, but some of them address a small amount of information…
“21)I feel there is a significant number of headings in the text and that difficult the perception of the work and results obtained
Author’s response:
We used no more than 3rd order of subheadings, especially in 2, and 3 sections, to help readers and to better distinguish different aspects of our work, to provide a higher level of order. Our work is multicontextual so we would like to use these subsections.”
Figure 1: check caption… “from 17.0 72.6 MPa.”
In my perspective, the Discussion section still needs to be improved. See that when references are used in the context of the discussion of the results obtained in the scope of a specific study, the contents must be clearly connected, and sometimes the “temptation” is to start writing like an introduction, and that connection is lost. I acknowledge that authors improved the text, and if they understand I am trying to improve the relevance of this section, they will perform additional changes, for this +art of the text, so important, to be cohesive, and their work to be citable.
The link is there in most cases, but reanalyse statements like this, why make the reference to that study then?
“Thus, it is difficult to compare Chinmayananda et al. [27] results with the present work.”
Leave spaces in Table 6, when presenting values for (kN) (really necessary two units?... and it must be corrected… “1 MPa = 106 N/m2 = 106/103 kN/m2 = 1,000 kN/m2”):
“3.43(0.405)-”
It is different… the start aims to justify the need for the study to be done, At the end, authors should defend findings
And see that the last sentence should not be bulleted…
“The technology of pelletization is well known”
Why bullets anyway? Connect it all, more difficult… Authors write bullets as if they were the “highlights” many journals request now… It makes it more difficult for the reader to link conclusions. It is my opinion.
“27)The conclusions section could start with a brief contextualization enhancing the study…
Author’s response:
We did that, but at the end of Conclusions.”
Add the DOI to more references.
Other changes: as done before in Table 4: the same repeated values could be in one merged cell, it helps the reader to focus on what alters…
“0.0”
“50.8”
Author Response
We would like to thank Reviewer #3 for significant comments. The manuscript has been improved. Details are in the attached file.
